# Assessing the Sustainability of Photodegradation and Photocatalysis for Wastewater Reuse in an Agricultural Resilience Context

Tiziana Crovella * and Annarita Paiano

Department of Economics, Management and Business Law, University of Bari Aldo Moro, Largo Abbazia Santa Scolastica, 53, 70124 Bari, Italy; annarita.paiano@uniba.it
* Correspondence: tiziana.crovella@uniba.it; Tel.: +39-320-01-88-148

**Abstract:** The growths in worldwide population—of up to 8.5 billion people by 2030—and agriculture have put great pressure on water resources, above all in arid and drought-prone areas. Nowadays, water scarcity, drought and pollution of wastewater are considered major issues of concern. For this reason, the authors provided an overview of two methods of wastewater purification and removing pollutants for use in crop irrigation in a sustainable manner. The novelty lies in the reuse of recovered wastewater, purified through photodegradation and photocatalysis technologies using solar energy. The knowledge of the environmental impacts associated with the use of recycled water with these photo-processes to irrigate crops under field conditions is still scarce. In the future, this issue will be important. In particular, photodegradation and photocatalysis achieve a sustainable reduction in contaminants contained in wastewater of between 35% and 100%. The use of bismuth vanadate supports the complete removal of pollutants, and the implementation of catalytic membranes makes these processes more circular. This research was performed under the "Progetto GRINS "Growing Resilient, Inclusive and Sustainable" with the aim of "Building a dataset for the circular economy of the main Italian production systems".

**Keywords:** photodegradation; photocatalysis; sustainability; circularity; wastewater reuse; agriculture; review

## 1. Introduction

Shortages of natural resources and growths in energy consumption and environmental pollution are growing problems worldwide [1,2]. In this context, water scarcity, which affects many regions in the world, has become a serious concern, so much so that it is included among the 2030 sustainability goals [3]. It is also greatly influenced by climate change, considering that the total annual man-made greenhouse gas emissions have increased by 70% from 1970 to today. Environmental pollution presents a serious danger to the environment and public health [4]. Considering that water pollution resulting from organic contaminants is a serious threat for the environment and human health [5,6], the removal of phenol compounds, pharmaceuticals and dyes is an essential practice [7,8]. It is extremely urgent to remove them from water with the systems available to the chemical and science industries [9]. The leakage of waste discharged into the ground leads to the contamination of groundwater; consequently, this requires the remediation of the contamination with the use of expensive processes to increase sources of water [10,11]. In particular, dangerous contaminants contained in reused wastewater, particularly metals and antibiotics, represent serious risks to human health [12].

For this reason, treating raw water through different purification technologies to reuse wastewater must be a widely recommended practice [3]. Considering that climate change, pollution and water scarcity are still unsolved problems, the reuse of treated wastewater is indispensable in countries suffering from drought [13]. Additionally, considering that the

knowledge of the environmental impacts associated with the photoprocessing of recycled water, such as photodegradation and photocatalysis, for use in irrigating crops under field conditions is still limited, it is necessary to provide new insights to researchers and stakeholders [14].

Photodegradation (Figure 1) is the degradation process of organic materials. As a result of the use of light radiation, or more generally electromagnetic radiation, it is considered an advanced green technology for solving environmental problems [15].

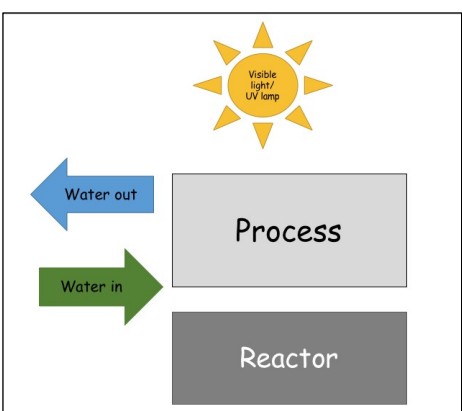

**Figure 1.** Photodegradation scheme. Source: personal elaboration by the authors.

Photocatalysis (Figure 2) is a natural phenomenon in which a substance, i.e., a photocatalyst (such as $TiO_2$ (titanium dioxide)), accelerates the rate of a chemical reaction through the action of natural or artificial light. Hence, photocatalysis is one of the best techniques for antibiotic degradation and an alternative to conventional chemical approaches. It is an economic method, highly efficient and environmentally friendly, with mild reactions and low impacts [16–18].

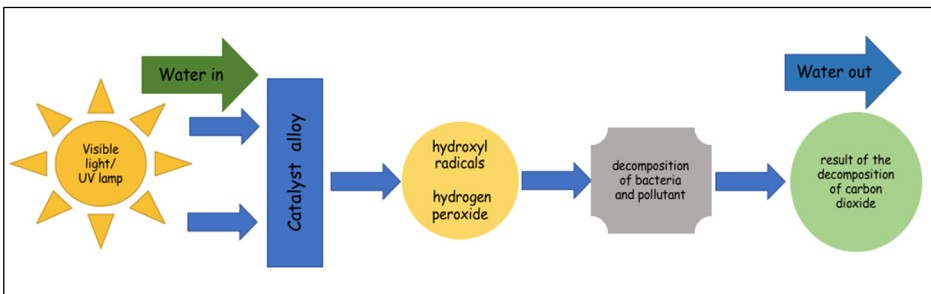

**Figure 2.** Photocatalysis scheme. Source: personal elaboration by the authors.

This paper uses photocatalysis technology, that has been successfully applied since 1972, when it was discovered, for its efficiency to solve energy crises and environmental contamination using solar energy, a natural green resource [19].

Practically, catalysis—the chemical phenomenon at the basis of photocatalysis—plays an important role in the degradation of organic contaminants, showing excellent photo-Fenton oxidation and organic contaminant decomposition activity (compounds such as phenol, 2,4-dibromophenol, 2,4,6-trichlorophenol, rhodamine B and methyl orange). These decomposition activities are necessary to clean wastewater to reuse it in agriculture [9]. Catalytic methods can efficiently stimulate peroxymonosulfate (PMS) through a non-photochemical path, and an elevated tetracycline (human health antibiotics polluting water) elimination of around 99.7%, can be achieved in 18 min [20].

Photocatalytic activity is increased by collecting more visible light with the aid of a Schottky diode, a metal–semiconductor junction with a low threshold voltage and a high switching speed, creating a tight connection between $CaIn_2S_4$ (graphite-like g-$C_3N_4$ hy-

bridized in CaIn$_2$S$_4$), reduced graphene oxide (RGO), Ti$_3$C$_2$Tx (two-dimensional titanium carbide) and MXene (two-dimensional transition metal carbide, nitride) [12]. Hence, this solution supports the use of solar energy to obtain a significant efficiency in the implementation of photocatalytic technology.

The effectiveness of these systems is proven by catalysts having a higher degradation adsorption capacity for TC (tetracycline) than other systems, a high stability and a universal applicability to degradation substrates. These systems reduce the stimulation of the energy barrier, support the relocation of bacterial and pollutant charges, improve the catalytic activity [20] and, therefore, have an excellent water purification capacity, as discussed above.

All these considerations support this study focusing on the applications of photocatalysis and photodegradation to underline the opportunity to use natural light, a renewable resource, to enhance sustainability and bio-circularity. Thus, we propose the use of solar energy as an effective solution to reduce energy consumption and reuse wastewater to avoid the global water crisis. In particular, solar energy is used for the treatment and purification of water for crop irrigation in agriculture.

In this context, this paper aims to develop a new theoretical framework for sustainable photodegradation and photocatalysis in order to provide a practical overview to support industries and future academic research in this field. With the purpose of addressing this aim, the authors have proposed nine research keywords (Table 1).

**Table 1.** Summary of the systematic literature review performed in this paper.

| No. | Keyword Combinations | Document Results |
|---|---|---|
| 1 | photodegradation AND photocatalysis AND wastewater AND reuse AND agriculture | 2 |
| 2 | photodegradation AND photocatalysis AND wastewater AND reuse AND LCA | 0 |
| 3 | photodegradation AND photocatalysis AND wastewater AND reuse AND water scarcity | 2 |
| 4 | photodegradation AND wastewater AND reuse AND agriculture | 9 |
| 5 | photodegradation AND wastewater AND reuse AND LCA | 1 |
| 6 | photodegradation AND wastewater AND reuse AND water scarcity | 9 |
| 7 | photocatalysis AND wastewater AND reuse AND agriculture | 7 |
| 8 | photocatalysis AND wastewater AND reuse AND LCA | 0 |
| 9 | photocatalysis AND wastewater AND reuse AND water scarcity | 6 |

Note: Source: personal elaboration by the authors.

Afterwards, we carried out a systematic literature review (SLR) to identify the variables that improve and facilitate wastewater reuse in agriculture, based on pollutant removal technologies.

This paper is presented in the Special Issue "Removal of Aqueous Emerging Contaminants through Photodegradation and (Photo)catalysis", with the aim of analyzing sustainable treatment technologies for contaminants, modelling their processes and providing practices for wastewater reuse in the agricultural context. Considering that some scholars have already presented the manufacture of new catalysts capable of effectively degrading pollutants through water treatment under all pedo-climatic conditions [20], the authors of this study have underlined the urgency of increasing the use of these technologies with the aim of stimulating greater sustainability and circularity in this sector.

In this study, a framework is proposed for enhancing the opportunities and strategies to implement the sustainable development and circular reuse of wastewater using natural resources, such as solar energy, in treatment and purification processes. Additionally, the novelty of this study is in stimulating the interest in research on the reuse in agriculture of contaminated water purified by photocatalysis and photodegradation technologies. To date, several studies have focused on the applications of purification systems, but less have focused on subsequent applications of the purified water [21]. Indeed, solar-powered semiconductor-based pollutant degradation has been receiving a lot of attention, but this is limited to the process, and less attention is focused on the applications [22]. These systems can result in greater efficiency of water resource utilization to resolve the challenges related

to water pollution and energy crises. There are already several policies to develop highly efficient C-doped polymeric carbon nitride (CPCN) catalysis, founded on the combination of morphological controls and in situ C-doping with a high degradation rate [21].

Finally, this research provides valuable insights for companies, academics experts and stakeholders, because it represents one of the few studies investigating the sustainability of water reuse in the agricultural sector.

## 2. Search Strategy and Theoretical Framework

Among the different ways of conducting a review of the scientific literature (for example qualitative, quantitative, critical, mixed, depth, systemic and systematic methods), the authors have conducted a quantitative systematic review of the literature in order to determine and evaluate the most important studies in this specific field by collecting and examining the relevant data. The PRISMA [23] methodology was used for identifying analysis clusters, methodologies, theoretical perspectives and common research areas [24]. Hence, this SLR is based on some theoretical postulations useful to identify the research path. In line with the explanation of photocatalysis and photodegradation [15–18], these two technologies symbolize green approaches which make use of various light sources (such as solar energy) and catalysts for the decomposition or degradation of emerging contaminants (including inorganic, organic, microbial and pharmaceutical contaminants) contained in wastewater. In this section, the authors have provided a snapshot of the research path: firstly, they have detailed the search strategy; secondly, they have determined the research to analyze from past review papers; and thirdly, they have applied an analytical approach based on the PRISMA Model. Particularly, the authors have compiled a complete sample of papers valuable for the critical analysis step.

The authors have adopted a clear, transparent and replicable strategy [25] to present an objective and effective analysis. Specifically, the authors have used nine combinations of the keywords indicated in Table 1 with the aim of building a review sample, with 2000–2023 as the timeline.

Subsequently, the authors have retrieved 36 papers from Scopus and Web of Science databases, following the same timeline, and through some filtering phases, they have built a snapshot of the current theoretical framework, analyzing the scientific literature on photodegradation and photocatalysis technologies. The general purpose is the collection of results on contaminant reduction.

The collected papers suggest possible solutions to address water scarcity, particularly the reuse of industrial wastewater after appropriate treatment using renewable energy sources. As matter of fact, it is possible to achieve a 35% reduction using 10 mg/L of substrate, 0.4 g/L of photocatalyst and 75kLux of solar radiation at a temperature of 31 °C [14] for several h (hours) of exposition. Generally, industrial wastewater containing hexavalent chromium can be treated with zinc oxide (ZnO), a semiconductor photocatalyst that under sunlight removes more toxic metals from its less toxic trivalent complex [14]. At the same time, it is also crucial to map the magnitude of the impacts following these processes. Specifically, a good approach to evaluate the environmental impact of UV-C (germicidal ultraviolet radiation)-based treatment systems for removing compounds from urban wastewater is an LCA (Life Cycle Assessment) [26]. This approach is an international standardized method that measures the possible impacts on the environment and human health associated with goods or a service in terms of the consumption of resources and emissions [27]. The LCA analysis has highlighted that adding hydrogen peroxide ($H_2O_2$) to the UV-C treatment drastically reduces the environmental impact of a photodegradation plant by means of a lower energy consumption [26].

Moreover, considering the concepts of the contaminant reduction percentage and environmental impact, the water scarcity, in terms of the accessibility of fresh water, adequate quality, lack of access to water services, and insufficiency due to the lack of adequate infrastructures, has also been considered [28]. Summarizing the results of previously published review papers on photodegradation and photocatalysis technologies, the authors underline

the necessity of studies dealing with photodegradation and photocatalysis as technologies to purify wastewater towards its efficient reuse, especially in agriculture.

Among the techniques previously analyzed, waste stabilization ponds (WSPs) remain an efficient and economical technology for wastewater management, especially in developing countries or in small regional areas. They are also adaptable to local contexts [13], such as in Italy. However, the performance of WSPs in the elimination of organic micropollutants is highly influenced by numerous elements such as their type, pond configuration, operating parameters, wastewater quality, the characteristics of the pollutant to be removed and, in particular, environmental factors (sunlight, temperature, pH) [13]. The mixing of supplementary wastewater treatment practices with WSPs for the removal of recalcitrant biological micropollutants should be contemplated. Over the years, improvements to the purification processes have been achieved, especially in photocatalytic processes using semiconductors such as bismuth vanadate (BiVO$_4$), which is very effective in the removal of aqueous pollutants. It is driven by sunlight, is corrosion resistant, is characterized by a low toxicity and achieves a rate of removal of between 35 and 100% [11].

Additionally, the degradation of phenolic compounds depends on various factors such as catalyst loading, light concentration, primary intensity of pollutants, pH (potential of hydrogen) and the type and intensity of oxidants. The pH has the greatest effect because it influences the photo-oxidation of the contaminants through ionizing the photocatalyst and altering external charges to different extents [3].

Generally, in order to decrease the energy utilization of photocatalysts, especially those used for phenolic compound degradation, it is necessary to switch from UV (ultraviolet) lamps to low-cost photocatalytic systems. These plants capture sunlight, use inexpensive photocatalysts and have a low-cost fabrication using lignin-based materials [3]. Thus, the adoption of these approaches generates more environmentally and economically sustainable processes towards an improved bio-circularity.

Although we did not find any methodology that analyzed environmental impacts, such as the LCA, a recent paper [3] summarized the functioning principles of photocatalysis in sophisticated catalytic oxidation processes, emphasizing this process as a sustainable method for decreasing the manufacture of pollutants. Methodologically, these three papers have not been processed in the SLR (Table 2), but according to the PRISMA model [23], they are general and bibliometric reviews.

In particular, a comprehensive review of the state-of-the-art WSP applications used for removing organic micropollutants from wastewater has been presented [13]. In a critical review, the authors have analyzed the synthetic procedures and photocatalytic applications of visible-light-driven BiVO$_4$ in photocatalysts in the last ten years. Subsequently, they have provided some findings underlining BiVO$_4$ as a possible substance for photocatalytic applications in water treatment and environmental remediation [11]. Finally, through a general review, the latest improvements to photocatalytic AOPs (advanced oxidation processes) for the reuse of industrial wastewater, in particular of phenolic compounds, have been assessed. After analyzing these three review articles and removing them from the list of analyzed papers, the authors have conducted an SLR with the aim of analyzing the photodegradation and photocatalysis technologies to provide an overview of the sustainability of wastewater reuse in agriculture. Particularly, the authors have focused on a cluster analysis, environmental impacts and the hypothetical reuse in the agricultural field, with the aims of outlining more circular synthetic practices.

**Table 2.** Summary of past review papers.

| Document Methodology and Difference from This Manuscript * | Contaminants | Topics Analyzed * | Results * | Ref. |
|---|---|---|---|---|
| Comprehensive review, not an SLR | Pharmaceuticals, personal care products and endocrine-disrupting compounds | Overview of WSP treatment and performance, removing organic micropollutants in WSPs, pharmaceuticals and personal care products in WSPs, pesticides in WSPS, perfluorinated compounds, other organic micropollutants. | Efficiency, environmental and economic sustainability, performance: variable and influenced by many factors | [13] |
| Critical review, not an SLR | Dye molecules and actual organic pollutants | Structure, preparation, other syntheses, standard organic pollutants (methylene blue, methyl orange, rhodamine B, actual organic pollutants). | Effective visible-light-driven photocatalyst with excellent properties: narrow bandgap, resistance to corrosion, low toxicity, good removal efficiency (35–100%) | [11] |
| Review, not an SLR | Phenolic compounds (epoxy resin manufacturing, textile and leather industry) | Mechanisms and components of photocatalysis, key parameters in photocatalytic degradation, dominant industrial phenolic pollutants, photocatalytic degradation of phenolic compounds. | Higher energy consumption which depends on the light source, catalyst activity, process capacity, water quality and organic pollutants chemical structure and concentrations. It details the cost-effectiveness of a photocatalytic technique with a 100% removal of antibiotic content, 86% removal of TOC and 79% removal of COD in 420 min. | [3] |

Note: Source: personal elaboration by the authors. * acronyms listed at the end of the paper.

*Analytical Approach*

In order to differentiate this review from previous ones (Table 1), the authors have organized a database on Excel Sheets, detailing authors' names, article titles, document categories (e.g., research, review articles, articles and technical procedures), author keywords, abstracts, author affiliations, editorial journals, year of publication, DOI (digital object identifier) and research areas. Through this metadata reported in a PRISMA flow chart, the authors have identified, selected and analyzed the records to guarantee their suitability for this SLR.

Particularly, as indicated in the flow chart contained in Figure 3, the authors have followed four steps:

(1) They have selected review principles linked to the purpose of the study;
(2) They have examined the search items, removing nine duplicates, three review papers, one chapter and one ineligible paper;
(3) They have built a filtered sample, considering 22 eligible papers, and have carried out a content analysis and interpretation of three review papers with the aim of underlining the future research directions.

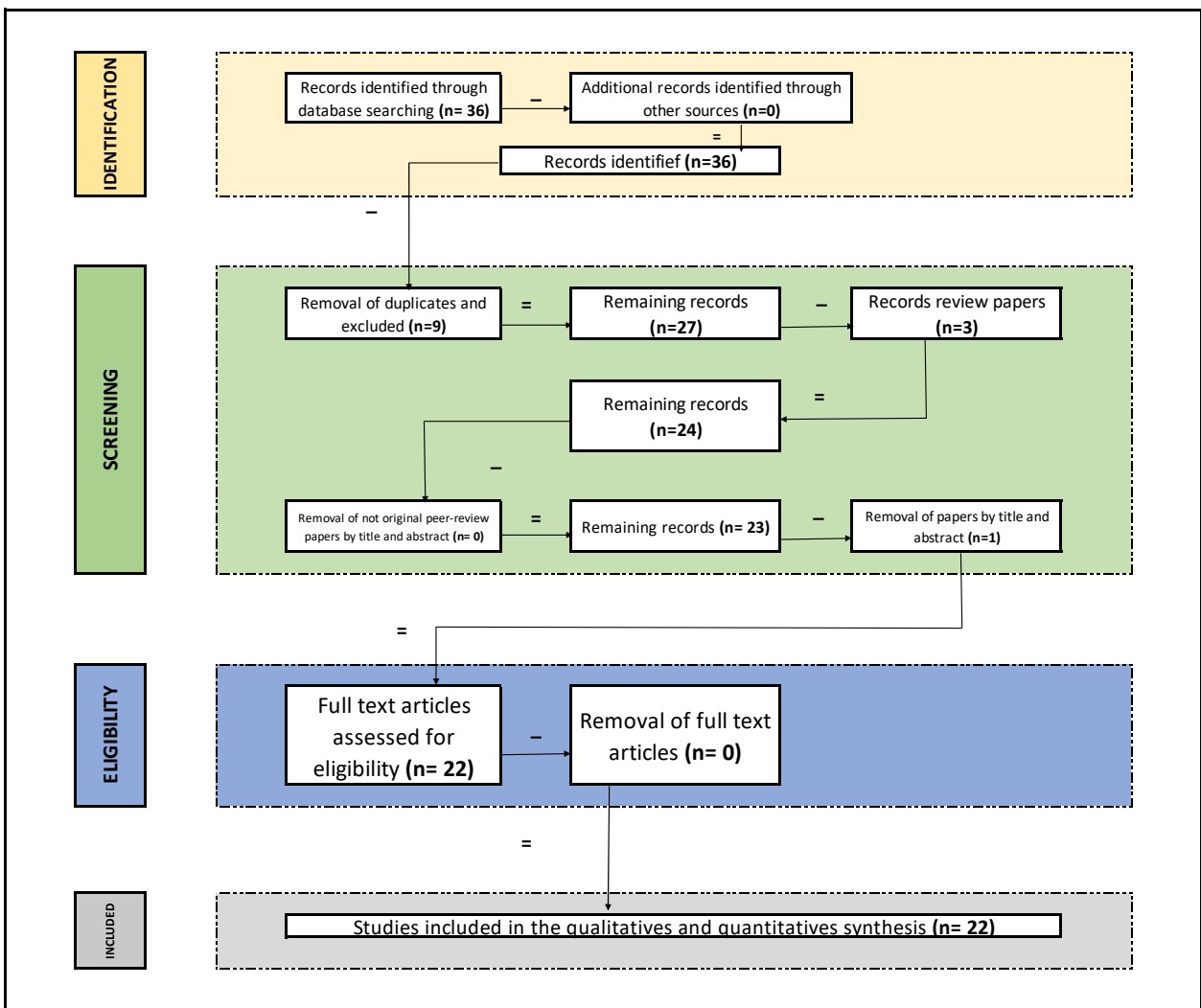

**Figure 3.** Flow chart of the SLR. Source: personal elaboration by the authors based on Page et al. [23].

Finally, the authors have critically analyzed the content of the 22 papers and provided a general overview of the purification of wastewater based on solar technology, supporting its reuse in an agricultural context.

## 3. Results

After the screening of the papers, it has to be stressed that past studies focused on solar energy consumption for the photoreduction and removal of pollution contained in wastewater [29]. Instead, recently, the environmental impact of these processes has been included in research, applying an LCA in order to evaluate the environmental impact of UV-C-based treatment systems for removing composites in urban wastewater [26]. Moreover, the processes of photodegradation and photocatalysis are often used to purify and remove micropollutants. Significant interest has been shown in pharmaceutical antibiotic removal from wastewater [30–32] and the removal of pesticides [14,33,34], insecticides [14] and bisphenol-containing microplastics and resins [35]. Wastewater also contains many micropollutants from pharmaceuticals and personal care products (shampoos, soaps, toothpaste and perfumes), including preservatives, masks and UV filters [36]. Similarly, at the municipal level, three emerging contaminants, i.e., sulfamethoxazole, pirimicarb and imidacloprid, must be removed from wastewater [37]. Little interest has been shown in the purification treatment, focusing on the risk to the environment and evaluating the indicators of the presence, persistence, bioaccumulation and toxicity of contaminants. Hence,

the impact of waste from the purification process has been evaluated [38]. Conversely, past studies have emphasized the efficacy of UV/chloramine for removal and observed the impacts of coexisting chloramines rather than the use of solar energy considered in this paper [39]. Moreover, the removal kinetics of organic chlorine, bromine and iodine contained in wastewater under solar radiation (not UV lamps) have been analyzed [40]. Finally, regarding the agro-industrial context covered by this paper, the treatment of olive mill wastewater in a photocatalytic process has also been studied [41].

*3.1. Sample Papers*

Recently, some scholars have proposed a methodology for identifying the indicators of the relevant contaminants of emerging concern during reuse for irrigation purposes [38]. It has emerged that the indicators that must be considered when reusing reclaimed wastewater for irrigation are soil and crop risk assessments, surface water and groundwater, which may come into contact with contaminants of emerging concern by surface runoff and percolation due to their movement once in the land. Notwithstanding the opportunity to assess several indicators, the LCA methodology represents the most complete and useful tool to carry out an analysis of UV-C photoreactors for the tertiary treatment of urban wastewater, determining the environmental impacts of different UV-C-based systems used to remove CECs (compounds of emerging concern) from MWWTP (Municipal Wastewater Treatment Plant) effluents [26]. Similarly, treated and gray water related to urban wastewater, which contains substances that absorb UV radiation or eliminate radicals, reduced the purification efficiency. Therefore, to overcome this problem, low pressure mercury atoms could be used, which have lower costs and would increase the purification effectiveness [36].

Qualitatively, for the reduction in irradiation times, the impact on costs and the environment and various parameters must be monitored. Particularly, it is essential to determine the effects on the products after the use of treated water. Mainly, in the case of treated wastewater used for crop irrigation, the same quality was recorded in lettuces grown using treated and non-treated wastewater [14,42]. It has been highlighted (Table 3) that natural solar photolysis reduces the concentration of total organic halogens below certain environmental conditions (pH, time of exposition and temperature) [14,42,43]. The combined use of renewable energy, such as solar irradiation, and persulfate (PS), a natural fertilizer, has given the process an additional advantage, especially in the Mediterranean, where many countries receive 3000 h of natural light per year [33]. In particular, the use of persulfate has eliminated insecticides in agro-wastewater in crop irrigation stages in a reasonable amount of time without reducing the product harvest or generating risks to human health [33]. Moreover, for the use of PS, the employment of zirconium oxide (ZrO$_2$), zinc oxide (ZnO) and ZrO$_2$/TiO$_2$ nanocomposites as a catalyst can be included in the treatment of mill effluents due to the high removal of pollutants of between 81 and 92% [41].

Even without the combination of other chemical or natural elements, the purification efficiency achieved under visible sunlight has reached up to 99–100% in removing antibiotic residues in domestic contexts and up to 62% in industrial plants [31]. However, the use of photocatalysts and ozonation catalysts in wastewater management has sped up the treatment of pollutants and increased the degradation of pesticides [34]. In particular, UV/chloramine is very effective for the purification of 1,4-dioxane (a heterocyclic organic compound) [39]. The removal of chloramines has increased the effectiveness of the treatment of organic pollutants, particularly those compounds that have a strong reactivity with hydrogen and oxygen. If chloramine is not easily removed, it is necessary to proceed to acidification with NHCl$_2$ (dichloramine) to facilitate the removal of pollutants [39]. Additionally, new polyoxometalate polymer composites are able to photodegrade diverse pollutants with several chemical configurations under specific sources of irradiation [35]. A titanium dioxide (TiO$_2$) paper/sunlight system was able to reduce 4-chlorophenol (4-CP) in the context of reusing wastewater for vegetable (tomato, onion and lettuce) irrigation [44].

Technically, between photocatalytic membranes and bare catalysts, the latter have superior activities due to the increased surface exposure of photocatalysts to dyes. However, bare catalysts and photocatalytic membranes have exhibited significant activity under solar radiation. Hence, contemplating the higher sustainability of photocatalytic membranes, which are reusable after a simple wash, photocatalytic membrane catalytic reactors present an interesting example of an integrated system in which molecular separation and chemical transformation take place in a single stage and at sustainable levels [45].

Notwithstanding the purification efficiency of these processes and the high pollutant removal rate, the potential risks to human and animal health associated with the increase in antibiotic resistance to purification processes must also be evaluated [32]. In particular, consideration must be given to the transport of antibiotics into soil systems via washing and runoff. The photocatalytic efficiency of the degradation of methylene blue under ultraviolet (UV) and natural sunlight reached up to 85% in a photocatalytic process [46]. Additionally, the efficiency of the photocatalytic activity can be improved by using purified clay for the mineralization of acridine orange [47]. Finally, in terms of economic sustainability, the greater cost of the photocatalyst compared to that of hydrogen peroxide may be mitigated by the reduced surface area required by sunlight [48].

**Table 3.** Sample papers analyzed after filtering.

| Cluster Analysis | Technological Approach | Findings * | Ref. |
|---|---|---|---|
| Experimental method, Municipal wastewater | Natural solar photolysis | Toxicity reduced from 12 to 50% | [43] |
| Municipal wastewater | Advanced oxidation process | Implementation of mercury lamps to increase the effectiveness of purification of municipal wastewater | [36] |
| Experimental method | Solar irradiation | No differences in quality of lettuce irrigated from treated water | [14] |
| Experimental method | Solar photocatalysis | No differences in quality of lettuce grown | [42] |
| Experimental method | NFC-doped photocatalytic oxidation | Efficacy of NFC-doped photocatalyst to remove antibiotics and pollutants | [31] |
| Experimental method | Photodegradation of bisphenol-A | Photocatalytic efficiency of new materials manufactured under different sources of radiation | [35] |
| Experimental method | Photocatalysis | Post-irradiation seed elongation of sprouts promotes the reuse of treated water | [44] |
| Experimental method | Solution combustion synthesis, Modified hydrothermal method | Photocatalytic membrane catalytic reactors are an interesting example of an integrated system in which molecular separation and chemical transformation take place in a single stage and at sustainable levels | [45] |
| Experimental method | Sorption, Desorption, Concentrations | Absorption and desorption of antibiotics vary according to the soil type and depend on pH, organic carbon, soil texture and cation exchange capacity | [32] |
| Experimental method | Photocatalytic activity | Good photocatalytic activity in the degradation of methylene blue using UV and visible light | [46] |
| Experimental method | Clay purification, Clay characterization, Adsorption experiments | Physicochemical characteristics of the treated effluent allow the use of wastewater for crop irrigation | [47] |
| Experimental method | Solar driven advanced oxidation processes | High-level solar technology with faster degradation kinetics observed for the sunlight process, offsetting the cost of the photocatalyst | [48] |
| Comparative method | Modified grafting, Process, Fixed concentration of cinnamic acid | Scaled-up reactors for the purification of agricultural wastewater | [40] |
| Experimental method | Raceway pond reactor pilot plant with two types of RPRs (raceway pond reactors) | Improvement in the treatment capacity and reduction in the consumption of chemical reagents such as hydrogen peroxide | [37] |
| Experimental method | Ag- and Zr-modified $TiO_2$ nanoparticles used in degradation | Phytotoxicity of the degradation of parents and pollutants to determine the environmental impact of treated water on three plant seeds | [30] |
| Experimental method | Novel Zr/Ag-$TiO_2$@rGO hybrid photocatalyst prepared | Analysis of degradation kinetics, products and further toxicity during photocatalysis | [49] |
| Comparative method | Comparison environmental profile of different UV-C-based systems | Major impacts due to electricity during the operating phase of the photoreactor | [26] |
| Experimental method | Chemical experiment | $NH_2Cl$ and $NHCl_2$ contribute to advanced oxidation processes (UV/AOPs) | [39] |
| Experimental method | Use of $TiO_2$ for photocatalysis | Simultaneous use of a catalyst increases the percentage of purification for the elimination of pesticides | [34] |
| Experimental method | Photooxidation of pollutant parameters from the OMW at different operational conditions | Incorporation of $ZrO_2$ into $TiO_2$ for a higher separation efficiency | [41] |
| Experimental method | Natural sunlight irradiation during the summer/autumn of 2017 | PS is effective, low cost and degrades pesticides in agro-wastewater in a reasonable amount of time and with solar energy | [33] |
| Methodological proposal | Identification of relevant indicators CEC for performance evaluation of new end-of-tube technology in a reuse project for irrigation purposes | Presents a shortlist of CECs included in regular monitoring programs during reuse operations | [38] |

Note: Source: Personal elaboration by the authors. * acronyms listed at the end of the paper.

In conclusion, the treatment of wastewater from agricultural industries can be defined as technically more complex, in particular for wastewater containing phenolic com-

pounds [40]. However, a high removal efficiency can be achieved, equal to about 80% of the micropollutants present in raceway reactors [37]. Furthermore, special consideration must be paid to the photodegradation processes and phytotoxicity of pharmaceutical antibiotics [30]. Among these, the purification and detoxification of p-bromophenol is achievable through a photocatalytic process under visible light [49].

*3.2. Bibliometric Analysis*

The earliest article was published by Elghniji et al. [44], and the last two were published in 2023. Scholars' interest in the subject has increased since 2019, with a peak of four articles reached. Figure 4 presents the research timeline.

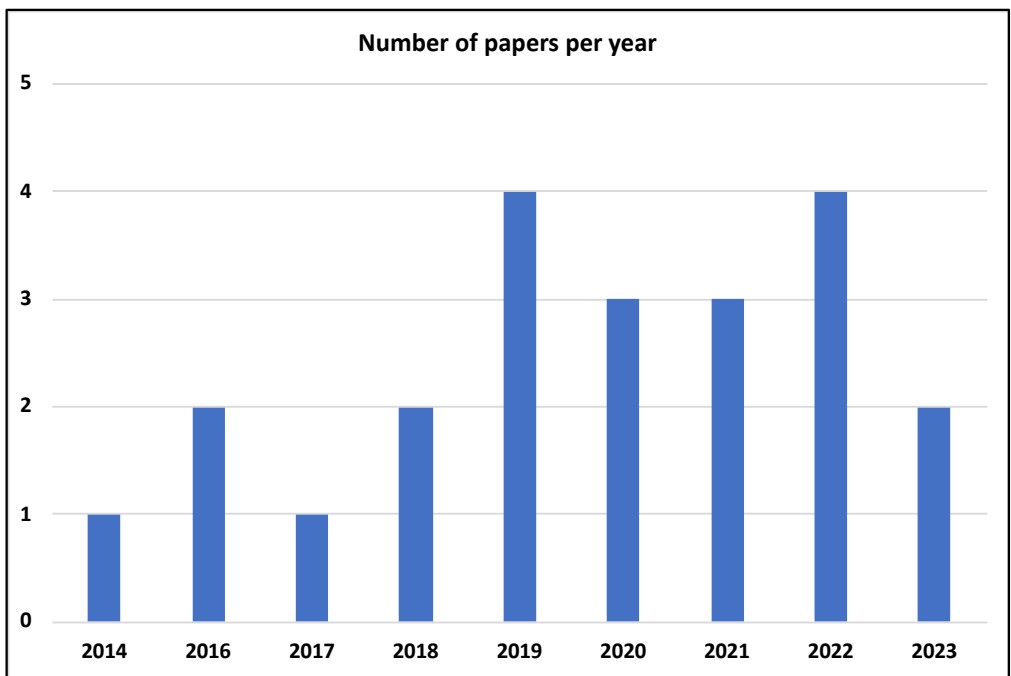

**Figure 4.** Number of papers published on photodegradation and photocatalysis topics per year. Source: personal elaboration by the authors.

Secondly, among the 16 journals in which the 22 selected papers were published (Table 3), *Science of the Total Environment presented four papers*, the *Journal of Environmental Management*, *Chemosphere and Ecotoxicology and Environmental Safety* presented two contributions each (Figure 5).

Most of the articles included experimental methods; additionally, two were based on a comparative analysis and one proposed a methodology. From a technical point of view, five papers focused mainly on photocatalysis and four on solar irradiation, five carried out experiments in pilot plants, three analyzed purification performances and five presented several advanced processes for the removal of contaminants.

In conclusion, the major contributions of the papers screened are useful for all stakeholders involved in agriculture, because they underline the capabilities of the purification of wastewater for circular reuse in crop irrigation. Specifically, the different methods applied can achieve rates of purification from 35 to 100% depending on the lighting conditions. For this reason, stakeholders could design an economically and environmentally sustainable system for photodegradation or photocatalysis.

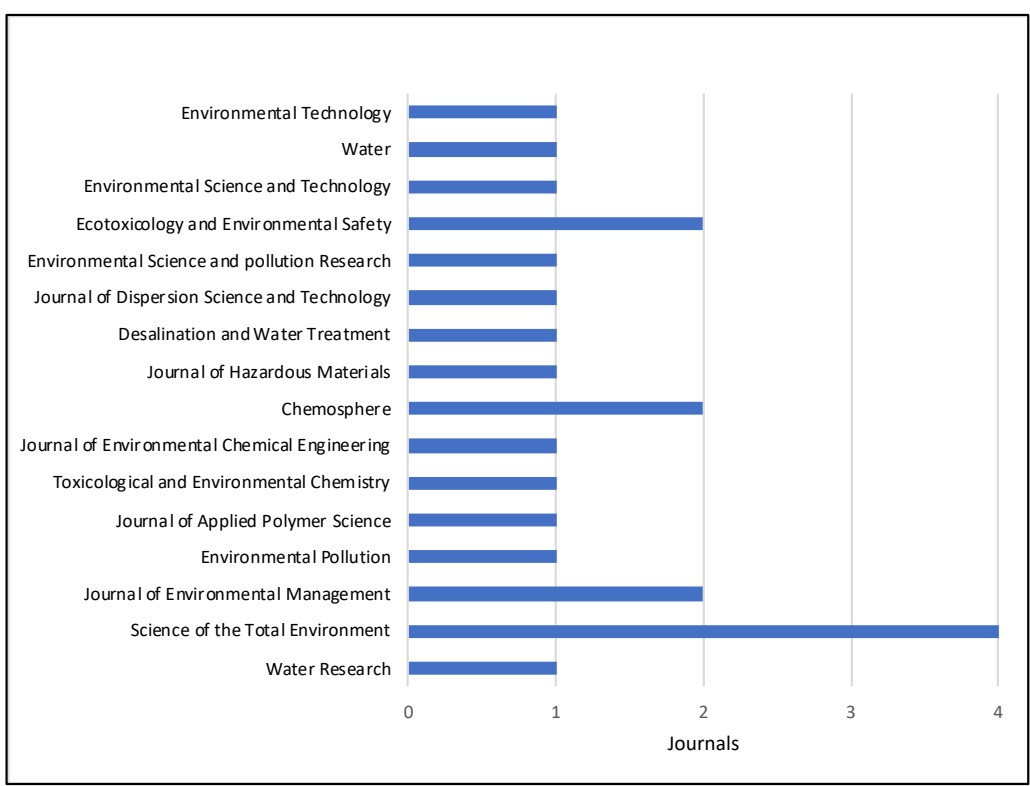

**Figure 5.** Number of articles on photodegradation and photocatalysis published per journal. Source: personal elaboration by the authors.

## 4. Discussion

Photolysis is the most used technology in recent years. Conversely, solar photocatalysis and the photo-Fenton process represent less implemented photocatalytic processes. In particular, photolysis represents the most environmentally friendly and economic process [33]. Photocatalysis and ozonation catalysis have been used frequently for pesticide removal [34]. However, as supported by some applications, the pollutant removal efficiency of photolysis can be increased by introducing hydrogen peroxide ($H_2O_2$) into the process [39]. Even in the case of pharmaceuticals, the photocatalysis approach is applicable; in particular, degradation and detoxification are efficient in the removal of p-bromophenol under visible light [30,49]. This approach can also operate in a clay purification context [47]. However, this technology is effective under sunlight and UV light; hence, it reduces lamp energy consumption is achievable using natural resources. In particular, methylene blue is able to degrade several contaminants under UV and visible light via photolysis [46]. Through other tests, the duration of photolysis can be minimized by reducing the use of direct sunlight, hydrolysis or other abiotic factors [32]. Moreover, some mixed-matrix membranes embedded in heterogeneous photocatalysts can reduce the cost of membranes [44,45]. These studies have shown that membranes are extremely efficient in the removal of toxic organic compounds, in particular dyes contained in wastewater. Additionally, photodegradation, tested in the context of cinnamic acid in natural sunlight and artificial UV light in agro-wastewater, has highlighted their efficiency [40]. Nevertheless, their feasibility has also been tested in a reactor under direct solar irradiance outside a university laboratory located in Southern Italy [48] or by applying photodegradation under UV-V and combined V-UV/UV-C irradiation [36]. This approach is also useful to improve the removal kinetics of total organic chlorine, bromine and iodine under sunlight [43] to rates similar to photocatalysis.

With the aim of reducing the environmental impacts, Verlicchi et al. [38] have proposed a membrane photoreactor for the removal of emerging contaminants and the reuse of wastewater. Notarnicola et al. [26] have combined photodegradation technology with the

LCA to analyze the environmental impacts throughout the pollutant removal process. In parallel, determination of the degradation mechanisms of pollutants and the simultaneous identification of photoproducts using mass spectroscopy is possible [35].

Among contaminant removal processes, photo-oxidation can be practiced at increasing photooxidation times (from 1 to 90 min) at diverse photocatalyst concentrations and different pHs (4.0–7.0–10.0) with lower UV irradiation, e.g., at 300 W (Watt) [41]. Another experiment was carried out with photocatalysts according to the sol–gel process without heat treatment [31]. The photo-Fenton process can be implemented for the removal of contaminants in urban wastewater treatment plants [37].

Generally, solar photocatalysis is the optimal advanced oxidation process (AOP) regarding the degradation of some pesticides and insecticide residues in agro-wastewater [14,42]. However, the extent of chemical removal that is necessary in the treated effluent in a WSP plant is related to the end use (natural waters or the beneficial and circular reuse for crop and park irrigation) [13]. Focusing on reuse proposals, analyzing the sample papers, some practical examples of wastewater reuse in the agricultural field have emerged. Particularly, the reuse of wastewater-polluted pesticides commonly used for lettuce [14], broccoli [33] and in all crop irrigation [30,42] was possible through the use of natural sunlight, titanium dioxide ($TiO_2$) and sodium peroxydisulfate ($Na_2S_2O_8$) [14]. In addition, there are no differences in the properties of lettuces irrigated by treated and non-treated wastewater. This water reuse can also be applied in polycarbonate greenhouses. Moreover, after adding advanced oxidation processes (AOPs), the elimination of pesticide residues from contaminated water is more efficient [42].

Generally, vegetables irrigated with irradiation systems exhibit a rapid increase in shoot length, particularly tomatoes, lettuces, onions and turnips [44]. This wastewater, including treated municipal water, is also useful for agricultural irrigation in basins [31,36] and forested soils [32].

In terms of environmental assessments, although the papers analyzed included some information about the process, the use of radiation, the synthesis adopted and the tests carried out, the authors found a lack of environmental impact information (Table 4). No papers included information about $CO_2$ emissions or GWP (global warming potential), and only two papers declared the amount of dissolved organic carbon (DOC). All papers provided the pH value at the end of the process. In Section 5, some authors underlined the environmental evidence recorded after process implementation and most of them indicated the treatment duration necessary to complete the removal of pollutants (Table 4).

The recycling of treated wastewater for irrigation or groundwater recharge has led to minor environmental impacts compared to other unconventional water supplies (e.g., water transfer or desalination) [14]. However, wastewater regeneration represents a solution for increasing water reserves in regions with water shortages and drought problems. These pollutant removal processes are critical to mitigating the environmental risks of pesticide use and toxicity and crop and soil effects [33]. Wastewater reuse was implemented to remove several micropollutants such as pesticides, antibiotics and other chemical residues by solar heterogeneous photocatalysis in agricultural crops, using more sustainable practices such as solar irradiation [14,42]. Considering that the risks to aquatic life, human health and crops are very high, the use of purification processes, such as photocatalysis and photodegradation, has reduced them and improved the water quality.

An evaluation of the environmental performance of the aforementioned methods is essential to evaluate whether the removal of pollutants from wastewater increases other environmental impacts. This insight deserves greater attention [26]. It has emerged that different exposure times are required for each type of pollution to be removed; for the strongest pollutant flecainide, an antiarrhythmic drug, a UV exposure of 120 min is necessary. Instead, the worst pollutant fluconazole required 30 min of treatment using UV-C + $H_2O_2$ [26]. At a medium level of exposure of 84 min, UV-C + $TiO_2$ removed the most persistent pollutant, i.e., lamotrigine, an antiepileptic drug used for the treatment of epilepsy [26].

**Table 4.** Results of environmental impact assessments of water quality after treatment *.

| GWP ($CO_2$ eq) | pH | DOC (m/L) | SUVA (m/L) | Br- (mg/L) | Nitrate (mg/L) | Turbidity (NTU) | $Cl_2$ TOX (mg/L) | $NH_2Cl$ TOX (mg/L) | Difficulties | Results | Ref. |
|---|---|---|---|---|---|---|---|---|---|---|---|
| x | 6–8 | 4.13 | 3.03 | 0.12 | 1.62 | 3.9 | 472 | 118 | Typical pH values in natural waters with limited influence on solar photolysis | Photolysis rates of removal: −50% of direct photolysis, Iodinated and brominated DBPs degraded faster than chlorinated DBPs under sunlight irradiation | [43] |
| x | x | x | x | x | x | x | x | x | Decentralized systems treating small volumes of water overcame the limited penetration of V-UV light into the water | In aqueous solution, photolysis was much faster when combined | [36] |
| x | 6–8 | x | x | x | x | x | x | x | Iron salts in Fenton homogeneous oxidation systems for negligible mass transfer limitations between catalysts and oxidants | Rate of oxidized pesticides > 99.5% | [14] |
| x | x | x | x | x | x | x | x | x | x | Rate of removal 90% | [14] |
| x | x | x | x | x | x | x | x | x | $TiO_2$ limited applications, with a 3.2 eV band gap energy | Purification yield between 62 and 99% | [31] |
| x | 6.5 | x | x | x | x | x | x | x | Very soluble clusters in aqueous solutions, limited recyclability and reuse as a photocatalyst, introduction of organic pollution to the treated environment | Degradation of the recalcitrant pollutant of between 50.3 and 100% in 90 min of irradiation with UV, solar and solar lamp and LED | [35] |
| x | 8.5 | x | x | x | x | x | x | x | Phosphomolybdic and decatungstate composites limited ability to adsorb (–18%) | Irradiation tests for 210 min produced minor and negligible toxicity, showing rapid increase in shoot lengths of tomato, onion and lettuce | [44] |
| x | 5–5.5 | x | x | x | x | x | x | x | Greater dispersion of photocatalysts in the aqueous system due to the limited availability of catalysts with little exposure to dyes in the photocatalytic process | Catalysts synthesized with exposure 1–51.71 min | [45] |
| x | 4.5–6.5 | x | x | x | x | x | x | x | Antibiotic absorption with plastic materials | Recovery rate between 15 and 81% of the compound | [32] |

Table 4. *Cont.*

| GWP (CO$_2$ eq) | pH | DOC (m/L) | SUVA (m/L) | Br- (mg/L) | Nitrate (mg/L) | Turbidity (NTU) | Cl$_2$ TOX (mg/L) | NH$_2$Cl TOX (mg/L) | Difficulties | Results | Ref. |
|---|---|---|---|---|---|---|---|---|---|---|---|
| x | x | x | x | x | x | x | x | x | x | 85% removal efficiency under visible light | [46] |
| x | 8 | x | x | x | x | x | x | x | High cost of adsorbent and its difficult regeneration | Removal efficiency rate of 80% in 60 min | [47] |
| x | 6.24–7.44 | x | x | x | x | x | x | x | Photocatalyst removal and poor absorption of semiconductor radiation under visible light | Complete removal in 96 h of treatment; between 27 min and 25 h, the removal rate was 93.5% | [48] |
| x | 3.8 | x | x | x | x | x | x | x | x | After 90 min of irradiation, effective photocatalysis for use in photodegradation | [40] |
| x | 5–5.5 | x | x | x | x | x | x | x | After 15 min, the final yield (−80%) was limited by photodegradation | Removal of more than 60% (5 min of exposition, 50% removal) | [37] |
| x | 3–9 | x | x | x | x | x | x | x | x | x | [30] |
| x | x | x | x | x | x | x | x | x | x | Rate removal 80.5% in 3 h | [49] |
| x | x | x | x | x | x | x | x | x | Elimination of antibiotics with different durations of UV exposure | According to the type of treatment, times of exposure (from 30 to 120 min) to UV rays for the removal of contaminants change | [26] |
| x | 5.8 | x | x | x | x | x | x | 2 | x | Removal rate: between 60 and 80% compared to 2 mM chloramine | [39] |
| x | 5–5.5 | x | x | x | x | x | x | x | Use of a higher UV intensity increased the degradation of imidacloprid | Complete removal (>99%) in 20 min with simultaneous catalysts, 30 min if only one photocatalyst is used. | [34] |
| x | 4.6 | x | x | x | x | x | x | x | Regeneration of the TiO$_2$–ZrO$_2$ nanocomposite is the crucial step for heterogeneous photocatalysis | 98% of pollutants photodegraded in 60 min of photooxidation at 21 °C and 300 W UV | [41] |
| x | 7.3 | 0.45 | x | x | x | x | x | x | Authorized reagent to treat the residue, avoid phytotoxicity problems in plants and prevent surface water contamination and filtration of PPP in underground soil | Rate of degradation of all pesticides was +90% | [33] |
| x | x | x | x | x | x | x | x | x | x | Rate of removal was between 20 and 80% | [38] |

Note: Source: personal elaboration by the authors. * acronyms listed at the end of the paper.

Therefore, to improve the degradation efficacy of photocatalysis, it is necessary to test several elements in the laboratory: the time of exposure, the characteristics of the photoreactor, the length and volumetric flow rate, the luminous intensity of the UV-C lamps and the stainless-steel mesh on which $TiO_2$ is immobilized [26].

In the future, it is desirable to use simple divisible and recyclable photocatalytic materials, encourage the use of sunlight and LEDs as cost-effective and sustainable light sources and develop non-toxic and cost-effective catalysts with a simple production method and efficient photon adsorptions [3]. Moreover, for reducing energy utilization and economic concerns (Figure 6), during the use of photocatalytic technology, the following should be considered:

(1) The selection of cost-effective photocatalytically active materials as waste residues in photocatalysis;

(2) The regeneration and recovery of photocatalytic materials;

(3) The exploitation of solar radiation or accessible light sources [3].

Moreover, it will also be necessary to design photocatalyst materials that are efficient under sunlight at ~5% UV radiation and ~43% visible light [50].

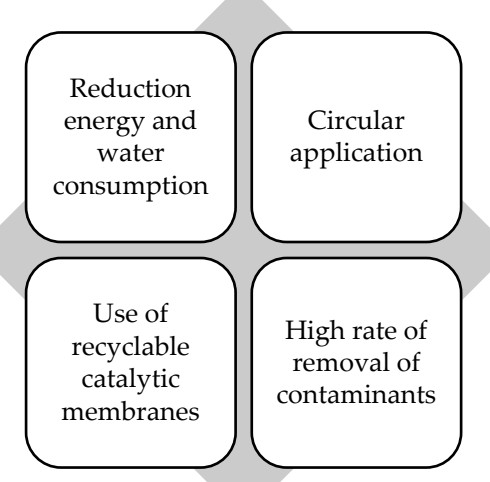

**Figure 6.** Positive aspects of the application of photocatalysis and photodegradation. Source: personal elaboration by the authors based on [51].

## 5. Conclusions

This review has included useful information for stimulating wastewater reuse in an agricultural context via treating water through photodegradation and photocatalysis under solar energy.

First of all, this review encourages scholars to analyze the pros and cons of the reuse of water treated with these two processes (photodegradation and photocatalysis) in agriculture. Several authors have already stated that there is no difference in the purification rate under lamp or solar energy rays; others have highlighted the growth of the crop germ. Other authors have declared that the use of photocatalytic membranes for the complete purification of water from organic micropollutants is a highly innovative and circular technology, as the membranes are reusable after a simple wash.

Considering that there are only few papers that have focused on photodegradation and photocatalysis for wastewater reuse in an agricultural resilience context, this literature review could stimulate scholars to investigate these technologies which are useful from a circular bio-economy perspective in agriculture. Particularly, state-of-the-art photodegradation and photocatalysis technologies in the literature have demonstrated the environmental

sustainability of these approaches and support the circularity of water, in particular for the reuse of wastewater in agriculture.

In turn, categorization of the articles (Tables 3 and 4) has delivered a conceptual framework useful for identifying important gaps in the scientific research and for suggesting links between the different aspects of wastewater reuse after photodegradation and photocatalysis, which will become progressively important in the simplification of these types of studies. Additionally, these results could offer practitioners and stakeholders an overview of the existing tools and strategies to improve photodegradation and photocatalysis technologies' environmental performance, especially in a resilient agricultural context. Particularly, the rate of purification (from 20 to 100% (Table 3)) depends on many factors that stakeholders can consider and overcome in order to build sustainable and efficient photocatalysis and photodegradation processes.

To summarize the main outcomes of this study, this review satisfies three main objectives:

1. Suggests the need for knowledge of sustainable best practices for stakeholders involved in agriculture;
2. Provides hypotheses for the reuse of wastewater purified with solar energy;
3. Elaborates on hypotheses of circular bioeconomy to best achieve the sustainability objectives of the 2030 Agenda.

Conversely, no mention has been included of the SDGs associated with water resources, considering their importance for the planet and water scarcity.

Therefore, to overcome the problems of the scarcity of water resources, the continuous need for agricultural products for human nutrition and growing environmental pollution, it is possible to use photodegradation and photocatalysis technologies to clean wastewater for crop irrigation and lead the agricultural industry towards bio-circularity [4].

In conclusion, this paper supports the sustainable development of agriculture as a key sector for guaranteeing global food security and the sustained and efficient use of water resources, which are key issues for the sustainable development of agricultural systems.

## 6. Future Directions and Limitations

Future research lines need to investigate photodegradation and photocatalysis treatments to reduce the potential health and environmental risks from the discharge of treated wastewater containing drugs; support wastewater reuse in agriculture considering the rate of purification; and capitalize on the reusability of photocatalytic membranes by including them in the process.

In summary, among the limitations, the environmental impacts that were avoided as a result of photocatalytic processes were not indicated in almost all papers. This information is useful for understanding the benefits obtained through the implementation of photodegradation and photocatalysis technologies, which already use a natural source, i.e., solar energy. Most papers did not present comparisons between reuse hypotheses; however, in some cases, there were a few examples, especially for the reuse in agriculture.

In the future, researchers, with the aim of improving the treatment of water to be able to reuse it, could include epidemiology analyses based on wastewater. This would present an additional advantage for a greater understanding of the possible resistance to the processes of photodegradation and photocatalysis on the microbial level.

**Author Contributions:** Conceptualization, T.C. and A.P.; methodology, T.C.; software, T.C.; validation, T.C. and A.P.; formal analysis, T.C.; investigation, T.C.; resources, T.C.; data curation, T.C.; writing—original draft preparation, T.C.; writing—review and editing, T.C.; visualization, T.C.; supervision, T.C. and A.P.; project administration, A.P.; funding acquisition, A.P. All authors have read and agreed to the published version of the manuscript.

**Funding:** This research was funded by "Progetto GRINS "Growing Resilient, Inclusive and Sustainable" cod. Id. PE0000018—CUP H93C22000650001—Finanziato nell'ambito del PNRR—Missione

4—Componente 2—Investimento 1.3—Unione Europea—Next Generation EU" with the aim of "Building of dataset for the circular economy of the main Italian production systems", Spoke 1.

**Data Availability Statement:** Not applicable.

**Acknowledgments:** This research was performed under "Progetto GRINS "Growing Resilient, Inclusive and Sustainable" cod. Id. PE0000018—CUP H93C22000650001—Finanziato nell'ambito del PNRR—Missione 4—Componente 2—Investimento 1.3—Unione Europea—Next Generation EU". Department of Economics, Management and Business Law, University of Bari Aldo Moro—Spoke 1, Code PNRR_PE_71, Thematic: Building of datasets for the circular economy of the main Italian production systems.

**Conflicts of Interest:** The authors declare no conflict of interest.

## Abbreviations

| | |
|---|---|
| 4-CP | 4 chlorophenol |
| AOPs | advanced oxidation processes |
| BiVo4 | bismuth vanadate |
| Br | bromide |
| $CaIn_2S_4$ | series of graphite-like $g$-$C_3N_4$ hybridized in $CaIn_2S_4$ |
| CECs | compounds of emerging concern |
| $Cl_2$ TOX | total organic halogen chloride |
| $CO_2$ | carbon dioxide |
| COD | chemical oxygen demand |
| CPCN | C-doped polymeric carbon nitride catalysis |
| DOC | dissolved organic carbon |
| GWP | global warming potential |
| LCA | Life Cycle Assessment |
| Lx | Lux, SI unit of measurement for illuminance |
| Min | minutes |
| MWWTP | Municipal Wastewater Treatment Plant |
| MXene | two-dimensional transition metal carbide, nitride |
| $Na_2S_2O_8$ | sodium peroxydisulfate |
| $NH_2Cl$ TOX | chloramine total toxicity |
| $NHCl_2$ | dichloramine |
| pH | potential of hydrogen |
| PRISMA | Preferred Reporting Items for Systematic Review and Meta-Analysis |
| PS | persulfate |
| RPRs | raceway pond reactors |
| SUVA | specific ultraviolet absorbance |
| TC | tetracycline |
| $TiO_2$ | titanium dioxide |
| $TiO_2$ | titanium dioxide |
| TOC | total organic carbon |
| TOX | total organic halogen |
| UV-C | germicidal ultraviolet radiation |
| UV/AOPs | ultraviolet advanced oxidation processes |
| $UV/H_2O_2$ | ultraviolet hydrogen peroxide |
| WSPs | waste stabilization ponds |
| ZnO | zinc oxide |
| $ZrO_2$ | zirconium oxide |

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
