# Peer review of "Assessing the Sustainability of Photodegradation and Photocatalysis for Wastewater Reuse in an Agricultural Resilience Context"

_water, doi:10.3390/w15152758_

Round 1

Reviewer 1 Report (Previous Reviewer 1)

The author has revised the comments I raised, so I suggest that the current version is acceptable.

Author Response

Reviewer 2 Report (Previous Reviewer 2)

This is a greatly improved version of the original paper, the content is now much clearer, but the English still needs to be revised some more. Some specific notes are below and in the accompanying file. Figures and tables are of good quality. References are relevant.

1.       Not every sentence needs to start with a furthermore, moreover, conversely, or therefore. These terms need to be used sparingly. Please write one of such terms once in every 5 (or more) sentences.

2.       Avoid one-sentence paragraphs.

3.       Do not make a first paragraph a summary of what is coming in that section, the subtitle does that already.

4.       Check for typos, a few are marked in the marked text.

5.       Terminology for wastewater treatment is wrongly applied, please revise.

see previous comment

Author Response

This manuscript is a resubmission of an earlier submission. The following is a list of the peer review reports and author responses from that submission.

Round 1

Reviewer 1 Report

This manuscript focused on methodological and environmental aspects, provided ideas on the practical applications of wastewater reuse in agriculture, identified possible future interests for photo technologies, provided an overview of the current reuse of wastewater, promoting circular bio-economy. However, there are still some problems should been addressed before publication. Thus, I recommend the work to be published in this journal after "major revised.

Below are my concerns:

1. The introduction should state the novelty of this study and progress of photodegradation and photocatalysis for wastewater reuse in agricultural resilience context. Some recent reviews could be helpful to you, such as synthesis and modification of ultrathin g-C3N4 for photocatalytic energy and environmental applications; Near-Infrared Light Responsive TiO2 for Efficient Solar Energy Utilization.

2. There are too many paragraphs with litter information in the introduction, which make the article seem not deep enough, the analysis is not thorough enough.

3. There are only a few papers focusing on photodegradation and photocatalysis for wastewater reuse in agricultural resilience context. Thus, what is the significance of this review?

4. The logical structure of this review could be enhanced. More discussion and summary should be added. These two articles can serve as templates: Strategies to extend near-infrared light harvest of polymer carbon nitride photocatalysts; 2D single- and few-layered MXenes: synthesis, applications and perspectives.

5. The format of the paper needs to be standardized, such as first line indent, upper and lower case, upper and lower script, etc.

6. What is the differences between this manuscript with the past review papers in Table 1?

7. The English of the manuscript should be polished carefully when you revise your manuscript.

8. What is the meaning of Table 3? Litter information can be obtained from this table.

The English of the manuscript should be polished carefully when you revise your manuscript.

Reviewer 2 Report

The topic is interesting but the manuscript does not follow the conventional guidelines of a technical paper. The manuscript seems to include everything the authors were able to find on the topic, without synthesizing and organizing it. There are too many non-technical descriptions about self-evident steps, for example describing what the authors will be including (a sentence) in the results and in the discussion (a sentence) sections, instead of just providing the results and the discussion. When writing about the particular contributions of studies, the authors list the author and then describe what that particular author found, then the next study, name the author and what that study found in particular, and so on.  The sentences are therefore disconnected, and there is too much unnecessary (not relevant) information that distracts the reader. Tables include too much information instead of the most important findings. The authors should check other review papers and use them as a guideline. Attached is a file with a FEW markings.

Style needs to follow conventional (non-technical) review articles; the present form is unacceptable. The grammar needs to be reviewed and there are a significant number of typos.

Reviewer 3 Report

The article listed the present literature photodegradation and photocatalysis tecnhologies and underlined their environmental sustainability in particular for wastewater reuse in agriculture. This work is suitable for Water and if modify some parts, it would be of interest to researchers in the community.

Comment 1: Some recent reated literatures are suggested to be mentioned in order to highlight the novelty of this manuscript, such as Chem. Eng. J., 2022, 438: 135623; Chem. Eng. J., 2021,421, 127838; J. Colloid Interf. Sci., 2022, 625, 466–478; Appl. Surf. Sci., 2021, 555: 149677.

Comment 2: There are few grammatical mistakes, please check the manuscript for grammar and English. All abbreviations should be defined at first mention in the manuscript, such as PRISMA.

Comment 3: Please ues Tables with three lines.

There are few grammatical mistakes, please check the manuscript for grammar and English. All abbreviations should be defined at first mention in the manuscript, such as PRISMA.
